# Analysis of Roadkill on the Korean Expressways from 2004 to 2019

**DOI:** 10.3390/ijerph181910252

**Published:** 2021-09-29

**Authors:** Minkyung Kim, Hyomin Park, Sangdon Lee

**Affiliations:** Department of Environmental Science and Engineering, College of Engineering, Ewha Womans University, Seoul 120-750, Korea; enviecol@ewha.ac.kr (M.K.); hyomin@ewhain.net (H.P.)

**Keywords:** wildlife vehicle collision, road-kill, reduction measure, time series, Mann–Kendall trend

## Abstract

Expressways in Korea are high-speed traffic roads connecting important cities. Road infrastructure continues to expand to accommodate the increase in traffic volume associated with the growth of industry and tourism. Here, data on 36,863 roadkill events that occurred on expressway routes managed by the Korea Expressway Corporation between 2004 and 2019 were analyzed. Characterizing patterns of roadkill is important for prioritizing roadkill mitigation measures. We identified consistently increasing or decreasing trends using Mann–Kendall statistics and Sen’s slope. Roadkill was most common in Gangwon Province and was concentrated between May and June and between October and December. Water deer (*Hydropotes inermis*) was the most common road-killed species. The trend analysis revealed a statistically significant decline in Gangwon Province and a statistically significant increase in the Capital Area and Chungnam Province. There was a significant increase in wild boar (*Sus scrofa*) roadkill in the first and fourth quarters. Mitigation measures are needed for regions and species showing increasing trends, including water deer in the first to third quarters, periods for which no decline in water deer roadkill was noted.

## 1. Introduction

Roads are a major cause of animal mortality worldwide [1]. The length of roads in Korea has continually increased since construction was first initiated, reaching 111,314 km in 2019 (www.stat.molit.go.kr, accessed on 22 September 2021). Expressways in Korea, which are major roads providing high-speed transportation exclusively for automobiles (www.ex.co.kr, accessed on 22 September 2021), connect important cities. A total of 4767 km of expressways were newly extended in 2019, and the length of these expressways continues to increase annually (www.stat.molit.go.kr, accessed on 22 September 2021). With the expansion of road infrastructure and the growth of the leisure and tourism industry, the traffic volume of expressways continues to increase. The construction of roads creates barriers to dispersal between habitats, and as more drivers use these roads, the number of animals killed by direct collisions with vehicles increases [2].

Studies of wildlife roadkill in Korea were first published in the early 2000s [3,4]. The Korea Expressway Corporation’s roadkill statistics for expressways were released in 2004; since then, roadkill has become a major social concern. Animal mortality on expressways has a major impact on ecosystems across Korea, including reducing biodiversity and damaging ecosystem health [5,6]. In addition, roadkill occurring on high-speed roads can even lead to large-scale vehicle accidents, and the resulting material damage can be quite large. Therefore, accurate roadkill data need to be obtained to aid the development of management and roadkill mitigation measures.

The Korea Expressway Corporation has been analyzing all mammalian roadkill accidents occurring on Korean expressways since 2004. Efforts are being made to reduce the frequency of roadkill on expressways, such as through the construction of fences that physically block the entry of animals onto roads, ecological pathways connecting wild animal habitats, and signs warning drivers of wildlife-crossing hotspots [7]. Despite these measures, animals continue to be killed on expressways.

The 15 years of roadkill data collected by the Korea Expressway Corporation from 2004 to 2019 provide important insights into patterns of road mortality on expressways in Korea. Analysis of temporal changes in roadkill frequency, along with the factors associated with roadkill accidents, could generate insights that could be used to prevent future accidents and help prioritize the implementation of different mitigation measures.

Here, patterns of roadkill on expressways in Korea were analyzed using this 15-year data set collected by the Korea Expressway Corporation. This analysis was used to develop and identify priority management targets. The results of our analysis of patterns of animal roadkill on expressways in Korea will aid the establishment of effective roadkill prevention measures.

## 2. Materials and Methods

The roadkill analyzed in this study was limited to routes (4151 km as of 2019) managed by the Korea Expressway Corporation and included high-speed national expressways defined in the Road Service Manual of the Ministry of Land, Infrastructure, and Transport. The roadkill data were collected by the Safety Patrols of the Korea Expressway Corporation. They recorded the number of roadkill events and details of the carcasses encountered during daily patrols of the expressway (three shifts a day). The location of expressway routes and the division of regions managed by the Korea Expressway Corporation are shown in Figure 1. Data on wild animal traffic accidents occurring on high-speed national expressways from 2004 to 2019 were provided by the Korea Expressway Corporation. We first examined annual changes in the 36,863 roadkill events recorded on the expressways, including patterns of roadkill by region, by species, and by month.

Next, we analyzed the occurrence of roadkill over time using time series roadkill data on expressways over the 15-year period. In this study, the seasonal Mann–Kendall Test was used to quantitatively analyze the pattern of monthly roadkill data with seasonality, and the Mann–Kendall test was used to assess the yearly pattern for each month. The Mann–Kendall test is a statistical method widely used to detect patterns; it is a nonparametric method used to detect increasing or decreasing trends [8,9]. Since this method was first proposed by Mann [10], the covariance matrix of Mann–Kendall statistics was developed by Dietz and Killeen [11], and this method can be applied to data showing seasonal fluctuations [12]. In addition, trends can be quantified by calculating the Sen’s slope, a non-parametric method that evaluates the slope of the data.

The statistic *S* can be obtained by Equation (1):(1)$$S=\overset{n-1}{\underset{k=1}{\sum }}\overset{n}{\underset{j=k+1}{\sum }}sign\left(x_{j}-x_{k}\right)\;\mathrm{where}:\mathrm{sign}\left(x_{j}-x_{k}\right)=\{\begin{array}{c} 1\;\;if\;x_{j}-x_{k}>0\\ 0\;\;if\;x_{j}-x_{k}=0\\ -1\;if\;x_{j}-x_{k}<0 \end{array}$$
and the test statistic Z is denoted by Equation (2).
(2)$$Z=\{\begin{array}{c} \frac{S-1}{\sqrt{VAR\left(S\right)}}\;if\;S>0\\ \;\;\;\;0\;\;\;\;\;if\;S=0\\ \frac{S+1}{\sqrt{VAR\left(S\right)}}\;if\;S<0 \end{array}$$

If Z > 0, the trend is increasing; if Z < 0, the trend is decreasing. Given a confidence level α, a statistically significant trend is observed if |Z| > Z(1-α/2), where Z(1-α/2) is the corresponding value of *p* = α/2, which follows the standard normal distribution. In this study, 0.05 confidence levels were used.

The magnitude of the time series trend was evaluated using a simple non-parametric procedure developed by Sen [13]. The trend was calculated by Equation (3), where β is the Sen’s slope estimate.
(3)$$\beta =\mathrm{Median}\left(\frac{x_{j}-x_{i}}{j-i}\right),\;j>i$$

## 3. Results

### 3.1. Roadkill Statistics from 2004 to 2019

#### 3.1.1. Temporal Variation in Roadkill

The total number of animal accidents that occurred on expressways nationwide from 2004 to 2019 was 36,863 (Table 1). The highest numbers of animal road accidents were observed in 2004 (2436), 2005 (3241), and 2007 (3216). From 2008 to 2015, roadkill frequency decreased in a gradual fluctuating manner; it then decreased after 2015, reaching a minimum in 2019 (1561). Despite the continued expansion of expressways, the number of roadkill events has decreased.

#### 3.1.2. Spatial Variation in Roadkill

Figure 2 shows the number and proportion of animal accidents occurring in different regions between 2004 and 2019 (Figure 2). The number of animal car accidents in Gangwon Province was over 7000 (19.2%); this was followed by Chungnam Province including Daejeon (17.8%), Chungbuk Province (16.3%), Jeonbuk Province (14.9%), Jeonnam Province including Gwangju (9.6%), Gyeongbuk Province including Daegu (9.2%), Gyeongnam Province including Busan (7.4%), and the Capital Area (5.6%).

#### 3.1.3. Interspecific Variation in Roadkill

Figure 3 shows the number and proportion of different species road-killed on expressways between 2004 and 2019. Out of 36,863 total roadkill events, water deer (*Hydropotes inermis*) accounted for 28,045 events (76.1%) (Figure 3). Water deer is the most common wild animal species in Korea, and water deer roadkill is common along roads throughout Korea, including national roads, expressways, and smaller rural roads [14]. The next most common road-killed animals were the raccoon dog (*Nyctereutes* *procyonoides*) (14.2%), the Korean hare (*Lepus* *coreanus*) (4.1%), wild boar (*Sus* *scrofa*) (2.0%), the Asian badger (*Meles* *leucurus*) (1.1%), the Siberian weasel (*Mustela* *sibirica*) (0.8%), and the Leopard cat (*Prionailurus* *bengalensis*) (0.7%).

#### 3.1.4. Monthly Variation in Roadkill

Roadkill was highest in May (7780, 21.1%), followed by June (5987, 16.2%), October (9.8%), December (8.0%), November (8.0%), April (7.7%), July (6.6%), September (5.5%), January (5.1%), August (4.7%), March (3.8%), and February (3.4%) (Figure 4). Approximately 37% of all roadkill accidents occurred on Korean expressways in May and June; approximately 26% of roadkill accidents occurred between October and December.

### 3.2. Analysis of Temporal Variation in Roadkill over 15 Years

#### 3.2.1. Patterns of Roadkill by Region

As the total number of roadkill events decreased over the 15-year period (Table 1), we expected that this would be accompanied by a decrease in roadkill across regions over this time period. However, the patterns of roadkill among regions varied (Figure 5). The seasonal Mann–Kendall analysis showed that there was a significant decrease in roadkill in most regions, but significant increases were noted in the Capital Area and Chungnam Province (Table 2). In addition, the Sen’s slope values of the Capital Area and Chungnam Province were 0.249 and 0.379, respectively, indicating that the rate of increase in Chungnam Province was greater (Table 2). This pattern may be related to local conditions such as the opening of new routes or the density of animal species; special roadkill prevention measures need to be implemented in these two areas where collisions with animals are increasing.

#### 3.2.2. Patterns of Roadkill by Species

Although the number of roadkill events decreased over the 15-year period, there was variation in the patterns of roadkill among species (Figure 6, Table 3). Significant reductions were observed in all animals, with the exception of water deer and wild boar. The highest slopes were observed for raccoon dogs (−3), Korean hares (−1), and Siberian weasels (−0.201). No significant pattern was noted for water deer. However, a significant increase in wild boar roadkill since 2011 was noted. There is thus a need for measures to be implemented to mitigate the increase in wild boar roadkill. Roadkill mitigation measures should also be implemented for water deer, which was the most frequently road-killed animal.

#### 3.2.3. Monthly Roadkill Patterns

The above analysis of the roadkill frequency by month revealed a clear monthly pattern. The pattern of animal accidents by month from 2004 to 2019 is shown in Figure 7. Clear decreases were observed in January, February, and from August to November; no patterns were observed for the other months (Table 4). The magnitude of the decline (i.e., the Sen’s slope value) was −19.350, −11.917, −9.573, −6.786, −5.833, and −4.663 for October, November, September, January, February, and August, respectively. The decrease in October was the most pronounced among all months. Thus, special mitigation measures need to be implemented for months in which roadkill has not decreased, including March, April, May, June, July, and December. Roadkill is highly concentrated in May and June (Figure 4); this, coupled with the fact that no decrease in roadkill frequency was noted for these months during the 15-year period, indicates that there is a pressing need to implement mitigation measures to reduce roadkill frequency in these months.

#### 3.2.4. Quarterly Roadkill Patterns

Seasonal changes in animal activity, especially movements such as dispersal after mating and childbirth, can affect seasonal roadkill patterns [15,16]. Consequently, we examined patterns of roadkill among species and quarters (Table 5). Overall, there was a significant decrease in roadkill in the first, third, and fourth quarters. This indicates that the second quarter is most in need of roadkill mitigation measures. A significant decrease in water deer roadkill was observed in the fourth quarter. Significant decreases in raccoon, wild rabbit, and weasel roadkill were observed in all quarters. There was a significant increase in wild boar roadkill in the first and fourth quarters. Thus, there was much variation in roadkill patterns among species and quarters. Mitigation measures are most needed for periods showing increases in roadkill and species for which roadkill frequency is not decreasing.

## 4. Discussion

Korea is continuously expanding its road infrastructure to accommodate the growth of the population and industry. Ungulate populations are also increasing in many countries because of changes in the number of large predators, hunting and wildlife management practices [17], and landscape structure [18,19,20]. As this is thought to be a general trend, the frequency of roadkill in Korea is likely to increase. However, fences have been installed on expressways in Korea to reduce the frequency of roadkill; as of 2020, fences have been installed in more than 50% of all expressways (Korea Expressway Corporation). The clear decrease in roadkill from 2004 to 2019 likely stems from the implementation of these mitigation measures, but the rate of decrease has recently slowed. Thus, reducing roadkill frequency further will require implementing measures that target priority periods, regions, and species.

As a result of roadkill statistics from 2004 to 2019, Gangwon Province with lots of forest region had the highest number of roadkill accidents, and water deer roadkill accounted for the largest proportion of roadkill accidents. In addition, May and June were the months during which most accidents occurred. According to Jakubas [21], that the highest number of mammal roadkill occurs in spring can be explained mainly by breeding activities, including roe deer parturition. Born in the previous year, baby deer wander irregularly to find new areas to abandon their family groups and settle in [22]. In the case of Korea, it is also considered that the first-year offspring of water deer, which is the species of the most roadkill occurred, leave their mothers and disperse during the period of May and June. This is consistent with the results of Kim et al. [23], who found that half of the roadkill that occurred in Korea in May and June were first-year-old male water deer.

Roadkill is caused by the close interaction of human and wildlife habitats worldwide [24]. Numerous accidents and various environmental impacts occurring globally make it difficult to predict roadkill. In order to analyze the causal relationship with roadkill, several factors such as traffic volume, land use, roads, and road infrastructure were studied [25,26,27]. In addition, since roadkill is an accident on the road, even the driver’s understanding of the road can affect the occurrence of an accident. However, a simpler and more intuitive forecast is needed to establish mitigation measures. This paper analyzed the 15-year time series of roadkill data to identify regions, periods, and species that did not show decreasing trends in roadkill frequency. These regions, periods, and species identified represent priority targets for future management.

The trend analysis indicated that there is a pressing need to implement roadkill mitigation measures in Chungnam Province and the Capital Area. In Gangwon Province, the region with the highest number of roadkill accidents, as well as other areas including this area, there was a clear decreasing trend; by contrast, in the Capital Area and Chungnam Province, there was a significantly increasing trend in roadkill. The current and future status of roadkill frequency needs to be assessed as mitigation measures are implemented.

Targeted mitigation measures need to be implemented for wild boar, as wild boar roadkill significantly increased from 2004 to 2019 in the first and fourth quarters. In addition, no significant decreases in water deer roadkill were observed in all quarters with the exception of the fourth quarter; thus, special mitigation measures need to focus on the first to third quarters for water deer. A significant decrease in roadkill was observed for all other species in all quarters.

Although other studies analyzing roadkill have been conducted on highways and national roads in Korea, no studies to date have analyzed roadkill patterns on a nationwide basis over a long period as examined in this study. Trend analysis provides important quantitative insights aiding the prioritization of roadkill management. In this paper, spatial heterogeneity, specific management regions, specific management species, and specific management periods were analyzed in light of species characteristics. The technique used in this study could be used to provide basic data to ensure the efficacy of environmental policies and aid future policy decisions; because non-experts can easily understand the magnitude of the observed changes, the results of this analysis can provide important scientific information that might aid roadkill management activities.

Future study of spatial and temporal patterns could provide key information for the management and mitigation of roadkill [28]. Analysis of the patterns of expressway roadkill by region and animal species, and their relationships with factors such as animal density and traffic density, could provide new insights that could reduce roadkill.

## 5. Conclusions

In conclusion, data on 36,863 roadkill events that occurred on expressway routes managed by the Korea Expressway Corporation from 2004 to 2019 were analyzed. A simpler and more intuitive forecast by characterizing patterns of roadkill is important for prioritizing roadkill mitigation measures. Priority management targets for reducing roadkill on Korean expressways based on the results of this study are presented below. First, measures need to be taken to reduce roadkill on expressways in priority areas (e.g., Capital Area and Chungnam Province) and species during apparently sensitive periods (e.g., wild boar in the first and fourth quarters and water deer in the first through third quarters).

## Figures and Tables

**Figure 1 ijerph-18-10252-f001:**
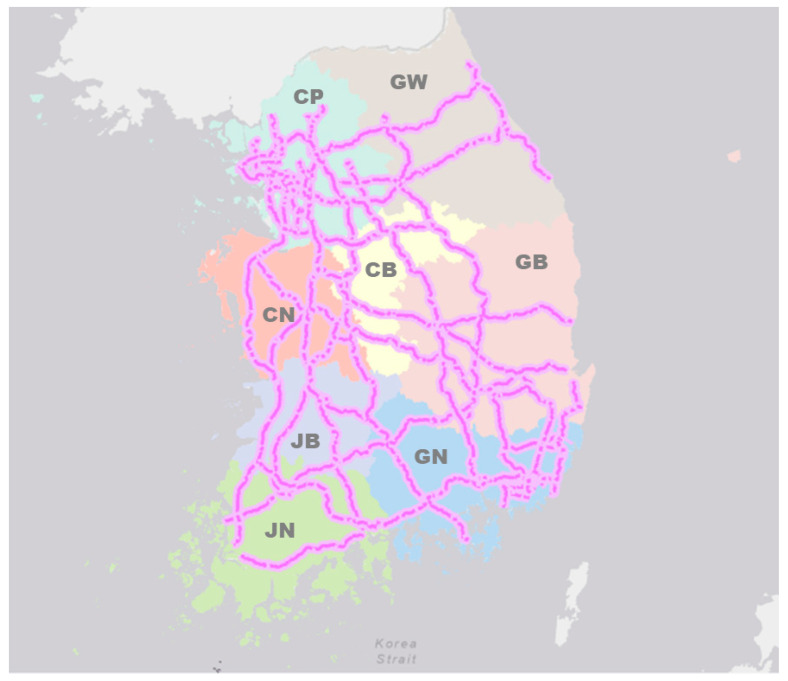
A map of the distribution of expressway and location of eight administrative districts in Korea where expressways are located. CP: Capital area, GW: Gangwon province, CN: Chungnam province, CB: Chungbuk province, JB: Jeonbuk province, GB: Gyeongbuk province, JN: Jeonnam province, GN: Gyeongnam province.

**Figure 2 ijerph-18-10252-f002:**
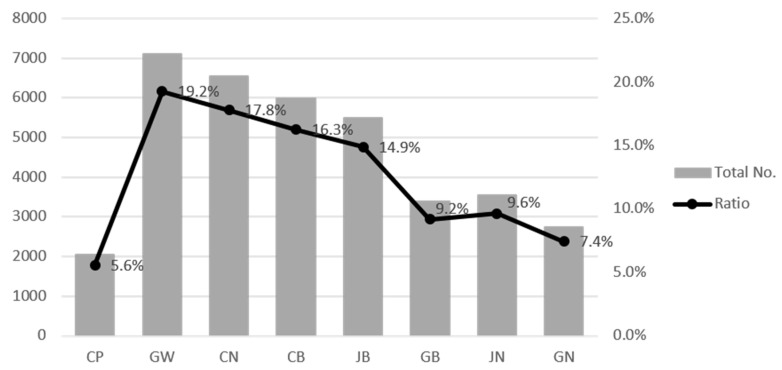
Total number and percentage (%) of roadkills by region on expressways in Korea between 2004–2019. (CP: Capital area, GW: Gangwon province, CN: Chungnam province, CB: Chungbuk province, JB: Jeonbuk province, GB: Gyeongbuk province, JN: Jeonnam province, GN: Gyeongnam province).

**Figure 3 ijerph-18-10252-f003:**
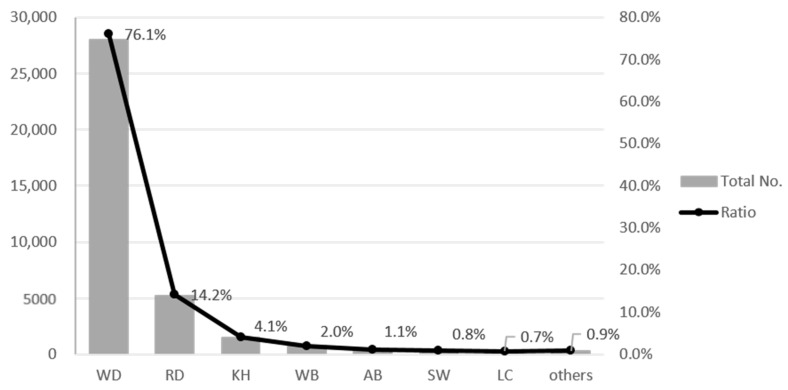
Total number and percentage (%) of roadkills by species on expressways in Korea in 2004–2019. (WD: Water deer, RD: raccoon dog, KH: Korean hare, WB: wild boar, AB: Asian badger, SW: Siberian weasel, LC: Leopard cat).

**Figure 4 ijerph-18-10252-f004:**
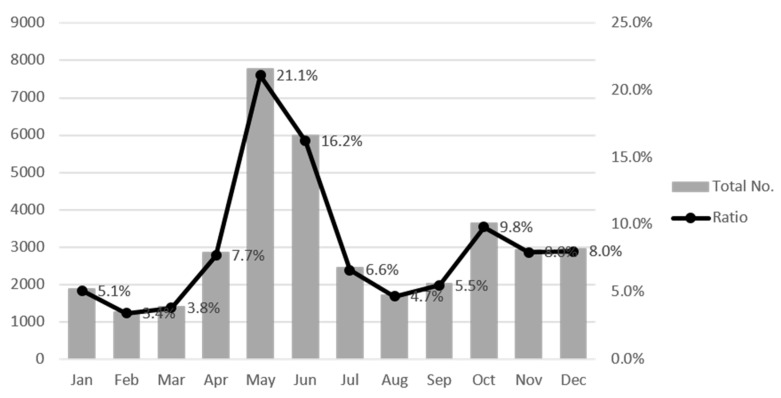
Total number and percentage (%) of roadkills by month on expressways in Korea in 2004–2019. (Jan: January, Feb: February, Mar: March, Apr: April, Jun: June, Jul: July, Aug: August, Sep: September, Oct: October, Nov: November, Dec: December).

**Figure 5 ijerph-18-10252-f005:**
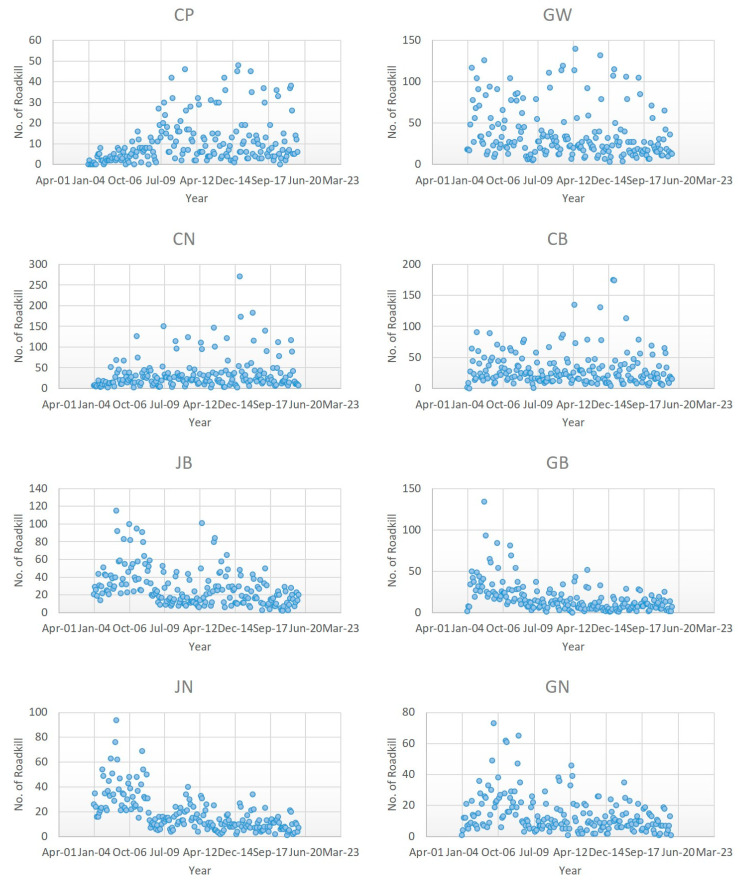
Distribution of expressway roadkills by region in 2004–2019 (CP: Capital area, GW: Gangwon province, CN: Chungnam province, CB: Chungbuk province, JB: Jeonbuk province, GB: Gyeongbuk province, JN: Jeonnam province, GN: Gyeongnam province).

**Figure 6 ijerph-18-10252-f006:**
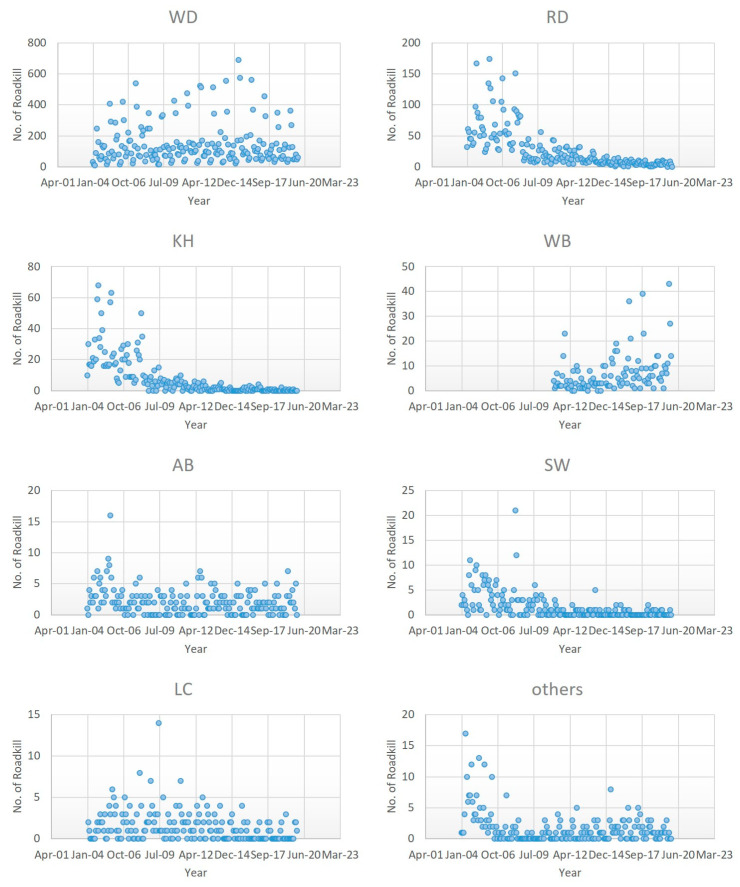
Distribution of expressway roadkills by species in 2004–2019(WD: Water deer, RD: Raccoon dog, KH: Korean Hare, WB: Wild boar, AB: Asian Badger, SW: Siberian Weasel, LC: Leopard Cat).

**Figure 7 ijerph-18-10252-f007:**
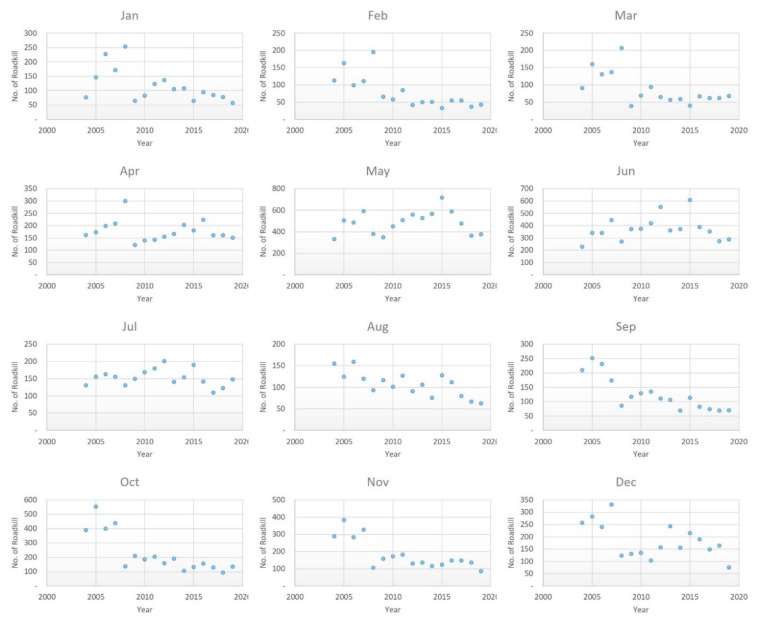
Distribution of expressway roadkills by month over 2004–2019.

**Table 1 ijerph-18-10252-t001:** 2004–19 Changes in the total number of roadkills by species on expressways in Korea.

Species	‘04	‘05	‘06	‘07	‘08	‘09	‘10	‘11	‘12	‘13	‘14	‘15	‘16	‘17	‘18	‘19	Total
** *Hydropotes inermis* **	1131	1779	1821	2167	1557	1490	1739	1914	1996	1939	1824	2302	1990	1643	1449	1304	28,045
** *Nyctereutes procyonoides* **	798	876	820	777	456	258	229	225	225	146	98	86	78	69	45	60	5246
** *Lepus coreanus* **	344	366	196	199	158	68	51	35	31	13	17	4	12	10	4	3	1511
** *Sus scrofa* **								68	43	31	48	96	115	115	90	143	749
** *Meles leucurus* **	37	69	24	21	23	17	14	17	26	28	20	17	22	19	18	29	401
** *Mustela sibirica* **	42	62	46	24	51	27	12	9	5	5	8	5	6		6	3	311
** *Prionailurus bengalensis* **	9	26	27	14	34	33	15	27	21	17	12	13	5	8	3	8	272
**others**	75	63	26	14	7	2	9	12	13	9	12	22	19	20	15	10	328
**Total**	2436	3241	2960	3216	2286	1895	2069	2307	2360	2188	2039	2545	2247	1884	1630	1560	36,863

(water deer [*Hydropotes inermis*], raccoon dog [*Nyctereutes* *procyonoides*], Korean hare [*Lepus* *coreanus*], wild boar [*Sus* *scrofa*], Asian badger [*Meles* *leucurus*], Siberian weasel [*Mustela* *sibirica*] and leopard cat [*Prionailurus* *bengalensis*]).

**Table 2 ijerph-18-10252-t002:** Seasonal Mann–Kendall trend analysis for each region.

Region	*p*-Value	S	Trend	Sen’s Slope
** CP **	** <0.0001 **	** 407.000 **	** upward **	** 0.249 **
**GW**	**<0.0001**	**−442.000**	**downward**	**−0.939**
** CN **	** 0.001 **	** 246.000 **	** upward **	** 0.379 **
**CB**	**0.030**	**−167.000**	**downward**	**−0.425**
**JB**	**<0.0001**	**−582.000**	**downward**	**−1.44**
**GB**	**<0.0001**	**−711.000**	**downward**	**−1.321**
**JN**	**<0.0001**	**−694.000**	**downward**	**−1.546**
**GN**	**<0.0001**	**−427.000**	**downward**	**−0.458**

Bold text are significant results, and red text indicates an increasing trend.

**Table 3 ijerph-18-10252-t003:** Seasonal Mann–Kendall trend analysis for each species.

Species	*p*-Value	S	Trend	Sen’s Slope
water deer	0.231	93.000	no trend	-
**raccoon dog**	**<0.0001**	**−1059.000**	**downward**	**−3**
**Korean hare**	**<0.0001**	**−963.000**	**downward**	**−1**
** wild boar **	** <0.0001 **	** 157.000 **	** upward **	** 0.561 **
**Asian Badger**	**0.014**	**−182.000**	**downward**	**0.000**
**Siberian Weasel**	**<0.0001**	**−617.000**	**downward**	**−0.201**
**Leopard Cat**	**<0.0001**	**−301.000**	**downward**	**0.000**
**Others**	**0.002**	**−222.000**	**downward**	**0.000**

Bold text are significant results, and red text indicates an increasing trend.

**Table 4 ijerph-18-10252-t004:** Mann–Kendall trend analysis for month.

Month	*p*-Value	S	Trend	Sen’s Slope
**January**	**0.043**	**−46.000**	**downward**	**−6.786**
**February**	**0.001**	**−73.000**	**downward**	**−5.833**
March	0.058	−43.000	no trend	-
April	0.928	−3.000	no trend	-
May	0.444	18.000	no trend	-
June	0.589	13.000	no trend	-
July	0.620	−12.000	no trend	-
**August**	**0.003**	**−66.000**	**downward**	**−4.663**
**September**	**0.000**	**−83.000**	**downward**	**−9.573**
**October**	**0.001**	**−78.000**	**downward**	**−19.350**
**November**	**0.010**	**−58.000**	**downward**	**−11.917**
December	0.163	−32.000	no trend	-

Bold text are significant results.

**Table 5 ijerph-18-10252-t005:** Mann–Kendall trend analysis for each species by quarter.

Species		*p*-Value	S	Trend	Slope
**All species**	**1Q**	**0.003**	**−66.000**	**downward**	**−16.938**
	2Q	0.620	12.000	no trend	-
	**3Q**	**0.001**	**−74.000**	**downward**	**−16.839**
	**4Q**	**0.002**	**−69.000**	**downward**	**−37.125**
**water deer**	1Q	0.224	28.000	no trend	-
	2Q	0.224	28.000	no trend	-
	3Q	0.652	11.000	no trend	-
	**4Q**	**0.012**	**−57.000**	**downward**	**−16.350**
**raccoon dog**	**1Q**	**<0.0001**	**−90.000**	**downward**	**−11.583**
	**2Q**	**<0.0001**	**−92.000**	**downward**	**−7.397**
	**3Q**	**<0.0001**	**−106.000**	**downward**	**−10.833**
	**4Q**	**<0.0001**	**−104.000**	**downward**	**−11.958**
**Korean hare**	**1Q**	**0.000**	**−87.000**	**downward**	**−4.522**
	**2Q**	**<0.0001**	**−92.000**	**downward**	**−2.528**
	**3Q**	**<0.0001**	**−97.000**	**downward**	**−2.000**
	**4Q**	**<0.0001**	**−99.000**	**downward**	**−4.231**
**wild boar**	** 1Q **	** 0.033 **	** 21.000 **	** upward **	** 0.838 **
	2Q	0.114	16.000	no trend	-
	3Q	0.118	16.000	no trend	-
	** 4Q **	** 0.048 **	** 20.000 **	** upward **	** 6.500 **
**Asian Badger**	1Q	0.236	−27.000	no trend	-
	2Q	0.172	31.000	no trend	-
	3Q	0.160	−32.000	no trend	-
	4Q	0.273	−25.000	no trend	-
**Siberian Weasel**	**1Q**	**0.002**	**−69.000**	**downward**	**−0.750**
	**2Q**	**0.000**	**−80.000**	**downward**	**−0.615**
	**3Q**	**0.002**	**−68.000**	**downward**	**−0.690**
	**4Q**	**0.001**	**−71.000**	**downward**	**−0.333**
**leopard Cat**	1Q	0.055	−43.000	no trend	-
	2Q	0.079	−39.000	no trend	-
	3Q	0.048	−44.000	no trend	-
	4Q	0.050	−44.000	no trend	-
**others**	1Q	0.151	−32.000	no trend	-
	2Q	0.277	−25.000	no trend	-
	3Q	0.489	16.000	no trend	-
	4Q	0.552	−14.000	no trend	-

Bold text are significant results, and red text indicates an increasing trend.

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
