# Peer review of "Analysis of Roadkill on the Korean Expressways from 2004 to 2019"

_ijerph, 2021, doi:10.3390/ijerph181910252_

Round 1
Reviewer 1 Report
No confidencials

Author Response
Response to Reviewer 1’s Comments:
Authors dealt with very interesting problem of road kills in S Korea Authors present interesting comprehensive study. Well done work in the context data curation. But msc. need some improvements especially discussion which is very poor. It makes that paper is not balanced (discussion is to short and not comprehensive).
Some remarks?
- Lines 225-226, lack of discussion related to months when deer road kills take place in others countries / biogeographical regions.
- Thanks for your comments. The review of this paper has been enriched through the papers below (Jakubas et al. 2018; Pagany 2020) that you recommended. As suggested by the reviewer, we added considerations for frequent months and regions. (line 224-235).
- It will be interesting to resolve (even based on small sample) if local or foreign drivers are more involved in road collisions. Some papers suggested that local drivers who know better their roads are involved in smaller number of road collisions compare to other ones.
- The topic you mention sounds very interesting. However, in this paper, we tried to present a simpler and more intuitive prediction in order to establish reduction measures. Therefore, citing other papers has been replaced by mentioning several other factors that may affect roadkill. (line 238-244).
- I also strongly recommend check format of paper.
- Conclusions have been added according to the format of IJERPH. References also follow the IJERPH format.
- I’m native speaker, probably paper needs supporting native speaker. But I understand this papers and English seemed to understandable for me.
- Yes, manuscript was edited by native English speakers and the grammar and content improved.
- Quality of paper will increase if authors discuss theirs results with data included in these papers:
Jakubas, D., Rys, M., & Lazarus, M. (2018). Factors affecting wildlife-vehicle collision on the expressway in a suburban area in northern Poland. North-Western Journal of Zoology, 14(1).
Pagany, R. (2020). Wildlife-vehicle collisions-Influencing factors, data collection and research methods. Biological Conservation, 251, 108758.
- As your opinion, our paper has been revised using two papers. We are very grateful for the improvement of the quality of our paper.
Reviewer 2 Report
This is a very interesting topics and the authors spent time and effort to analyze and organize all the raw data into a scientific research. Very well written both in English and style. Only few suggestion from me: Figure 2 & 3. It’s correct using bar chart to express difference between categories, but it’s not using line chart between categories (ratios of different areas and different animals). Figure 4. Sine the authors tried to express the monthly change of road kills, it is good enough to use the line chart; the bar chart seems redundant. Other than that, I am really looking forward to seeing how the results from this research can be applied into the field of highway construction and reduce the road kills.Author Response
Reviewer 2’s Comments:
This is a very interesting topics and the authors spent time and effort to analyze and organize all the raw data into a scientific research. Very well written both in English and style. Only few suggestion from me: Figure 2 & 3. It’s correct using bar chart to express difference between categories, but it’s not using line chart between categories (ratios of different areas and different animals). Figure 4. Sine the authors tried to express the monthly change of road kills, it is good enough to use the line chart; the bar chart seems redundant. Other than that, I am really looking forward to seeing how the results from this research can be applied into the field of highway construction and reduce the road kills..
- Thank you very much for your good comments. In the case of Figures 2 and 3, as in your opinion, bar figures are used to show the differences between categories. However, please understand that it was used to indicate information about the ratio of each region and each species of roadkill that occurred in Korea over the past 15 years. Likewise, it is intended to provide information about the figures that occurred each month in Figure 4. The ratios to total values are duplicated but tabulated together to aid readers who wish to ascertain probabilistic and statistic overview. Hope for your generous understanding.